# Risk and Clinical Significance of Idiopathic Preterm Birth in Microvillus Inclusion Disease

**DOI:** 10.3390/jcm10173935

**Published:** 2021-08-31

**Authors:** Changsen Leng, Yue Sun, Sven C. D. van IJzendoorn

**Affiliations:** 1Department of Biomedical Sciences of Cells and Systems, Centre for Liver, Digestive and Metabolic Disease, University of Groningen, University Medical Centre Groningen, 9713 AV Groningen, The Netherlands; lengchangsen@hotmail.com (C.L.); y.sun@umcg.nl (Y.S.); 2Department of Thoracic Surgery, Guangdong Esophageal Cancer Institute, State Key Laboratory of Oncology in South China, Collaborative Innovation Centre for Cancer Medicine, Sun Yat-sen University Cancer Centre, Guangzhou 510060, China

**Keywords:** microvillus inclusion disease, MYO5B, STX3, preterm birth, premature birth, congenital microvillus atrophy, congenital diarrheal disorder, intrahepatic cholestasis, PFIC

## Abstract

Microvillus inclusion disease (MVID) is a rare enteropathy caused by mutations in the *MYO5B* or *STX3* gene. MVID is a disease that is difficult to manage with clinical heterogeneity. Therefore, knowledge about factors influencing MVID morbidity and mortality is urgently needed. Triggered by a recent study that reported a high percentage of preterm births in twelve cases of MVID, we have conducted a comprehensive retrospective study involving 88 cases of MVID with reported gestational ages. We found that moderate to late preterm birth occurred in more than half of all cases, and this was particularly prominent in *MYO5B*-associated MVID. Preterm birth in MVID counterintuitively correlated with higher birth weight percentiles, and correlated with higher stool outputs and a significantly shorter average survival time. Data from this study thus demonstrate an increased risk of preterm birth in *MYO5B*-associated MVID, with a clinical impact on morbidity and mortality. Adverse effects associated with preterm birth should be taken into account in the care of children diagnosed with MVID. Documentation of gestational age may contribute to a better prognostic risk assessment in MVID.

## 1. Introduction

Microvillus inclusion disease (MVID) is a rare enteropathy [1,2]. MVID clinically presents with severe congenital intractable secretory diarrhea, malabsorption and failure to thrive [1,2]. MVID is an autosomal recessive disease associated with bi-allelic mutations in the *MYO5B* gene [3] or bi-allelic mutations in the *STX3* gene [4] in ~95% or ~5% of cases, respectively [5,6]. Symptoms of MVID are also present in some patients diagnosed with familial hemophagocytic lymphohistiocytosis caused by mutations in the *STXBP2* gene [5,7], but because of their distinctive hyper-inflammatory symptoms these patients are not diagnosed with MVID.

MVID is uniformly fatal if left untreated. Treatment is only supportive and involves life-long maintenance of nutrition and hydration with total parenteral nutrition (TPN). Prognosis is generally poor, and most patients die within one year after birth, mostly due to sepsis or liver disease [8]. With more case reports being published, it has become clear that the clinical presentation and course of MVID can be heterogeneous. Some patients could even be weaned off TPN [9,10,11]. Because MVID is a difficult to manage disease [12], more information is needed about factors that influence MVID morbidity and mortality.

Recently, Caralli and colleagues [13] noted a relatively high prevalence of preterm births in MVID when compared to other congenital diarrheal disorders. However, their report was based on only twelve cases. Worldwide, preterm birth-associated complications are the leading cause of death among young children [14]. Moreover, preterm birth is associated with gastrointestinal tract immaturity and growth failure, which are key features of MVID [1,15]. Therefore, more data are needed to confirm or refute the high prevalence of preterm birth in MVID, and, if confirmed, determine its significance for MVID morbidity and mortality.

## 2. Methods

### 2.1. Collection of Case Reports

To collect all published reports on cases of MVID, we searched EMBASE and MEDLINE databases using the following search strings: ((microvill* inclusion disease) OR (microvill* atrophy)) AND case report). Data from these cases were manually analyzed (using the ‘find’ tool in Adobe reader) with regard to preterm birth (defined as parturition prior to 37 weeks of gestation) [16], sex, birth weight, time of onset of diarrhea, stool volume, TPN dependency, neonatal respiratory distress, liver disease, polyhydramnios, gene mutations and survival (Appendix A).

### 2.2. Statistics

Odds ratios (OR) were calculated as measures of association. Statistical analyses included chi-square test with Yates correction to prevent the overestimation of statistical significance for small data, Pearson’s correlations and Student’s or Welch’s two-tailed t-tests to determine statistical significance, assuming a priori statistical significance when *p* < 0.05. The Mann–Whitney U test was used for non-normally distributed data sets. Levene’s test was used to determine homogeneity of variances. The Shapiro–Wilk test was used to determine normality. Sex-corrected birth weight percentiles were calculated at https://www.audipog.net/Courbes-morpho. Graphs were generated at https://canva.com/graphs and with MS Excel (2103 (16.0.13901.20400)/accessed on 13 April 2021).

Due to the retrospective nature of this study, data were missing in several variables because of incomplete data recording. Because our aim was to simply describe and compare subgroups of patients with different gestational ages (i.e., preterm versus term), no method for missing data was utilized.

## 3. Results

### 3.1. Increased Risk of Idiopathic Preterm Birth in MVID

In this retrospective study, study subjects were retrieved from a recently published collection of 131 published MVID cases since 1978 [8], complemented with 44 more recently reported cases [11] and one additional case from the literature search. From this collection of 176 MVID patients, gestational age was documented in 88 cases (Appendix A) and on average was 35.6 weeks. Preterm birth occurred in 52% (46/88) of these cases without sex preference. This is significantly higher when compared to the world average of 10% (OR 9.42, 95% CI 4.34–20.46, *p* < 0.0001). Of the preterm births in MVID, 91% (42/46) were categorized as moderate to late preterm birth (33^+0^–36^+6^ weeks gestation) (Figure 1A).

Of the preterm births in MVID for which the mode of delivery was documented, 84% (27/32) were spontaneous (that is, without elective labor induction or cesarean section). Medically indicated factors linked to late preterm birth, such as intrahepatic cholestasis during pregnancy, maternal diabetes, placental abruption, placenta previa, hypertension, preeclampsia, intrauterine growth restriction or multiple gestation, were not or rarely reported. Polyhydramnios (defined as an excess amount of amniotic fluid) was reported in 22% (12/55) of MIVD patients, for which the presence or absence of polyhydramnios was documented. However, there was no significant difference in the occurrence of polyhydramnios between preterm and term births in MVID (*X*^2^ (1, *N* = 54) = 0.0443, *p* = 0.83).

The average birth weight of MVID patients was significantly lower in preterm born patients (2584 ± 612 g versus 3191 ± 485 g (mean ± SD) for preterm and term births, respectively; *t* (62) = −4.36, *p* = 0.00005) (Figure 1B). Gestational age positively correlated with birth weight (Figure 1C). Notably, for 58% (19/33) of preterm MVID patients, the average birth weight was within the normal range (2500–4000 g regardless of gestational age). Moreover, when corrected for sex and gestation age, birth weight percentiles for preterm births were significantly higher when compared to term births (70.1 ± 19.9 and 48.3 ± 28.9 (mean ± SD), respectively; *t* (46) = 2.96, *p* = 0.005). Forty-one percent (12/29) of preterm birth weights in MVID were above the 80th centile (for term MVID cases this percentage was 12% (2/17)) (OR 5.29, 95% CI 1.02–27.57, *p* = 0.048) (Figure 1D), albeit still considered as appropriate for gestational age.

Together, MVID is associated with an increased risk of idiopathic preterm birth and a therewith associated higher average birth weight percentile.

### 3.2. Relationship between Preterm Birth and MVID-Causing Gene Mutations

We next investigated whether preterm birth in MVID was associated with specific causative genes. No MVID patients with *STX3* mutations were born preterm [4,17,18], whereas 76% (16/21) of MVID patients with *MYO5B* mutations were born preterm (Figure 2). Hence, for the MVID patients where the gene mutation had been identified, the increased risk of preterm birth in MVID was restricted to patients with *MYO5B* mutations.

Bi-allelic *MYO5B* mutations, excluding bi-allelic nonsense and/or frameshift mutations that give rise to premature termination codons and/or loss of the RAB11A binding site in the encoded myosin Vb protein, were also found in 14 patients who displayed no or episodal intestinal symptoms and were diagnosed with progressive familial intrahepatic cholestasis (PFIC)-type 6 [19,20]. Notably, none of these PFIC6 patients were born preterm [19,20].

These results indicate that preterm birth in MVID appeared restricted to patients with bi-allelic *MYO5B* mutations, but that not all bi-allelic *MYO5B* mutations increased the risk of preterm birth.

### 3.3. The Impact of Preterm Birth on MVID Morbidity and Mortality

Clinical characteristics of MVID, such as time of onset, extent of diarrhea, TPN dependency, intrahepatic cholestasis, neonatal respiratory distress and time of death display a poorly understood inter-patient heterogeneity [8,11,21]. We next addressed the relationship between these characteristics and preterm birth in MVID.

#### 3.3.1. Diarrhea and TPN Dependency

Preterm born individuals have an immature gastrointestinal tract. Immaturity of the small intestine is a key feature of MVID, and is associated with a secretory phenotype [15]. There was no difference in the time of onset of diarrhea between preterm and term births (*t* (70) = 0.364, *p* = 0.36), but a higher average stool output was observed in preterm births when compared to term births (166.5 ± 54 versus 126.26 ± 49.6 mL/kg/day in preterm versus term births, respectively; *t* (31) = −2.15, *p* = 0.02) (Figure 3A). All MVID patients required TPN, hence without correlation to gestational age at birth or birth weight percentile. The few reported patients that could be weaned off the TPN were all born at term [9,10]. Preterm birth in MVID thus may have influenced the extent of diarrhea and limited the chance to be weaned off TPN.

#### 3.3.2. Respiratory Distress

Neonatal respiratory distress has been reported in several cases of MVID, but how often it occurred in MVID has not been investigated. In our MVID case report collection, neonatal respiratory distress was reported in 16% (14/88) of cases of MVID, often preceding the onset of diarrhea and requiring assisted ventilation. In the remaining 74 cases, there was no mention of respiratory distress, meaning that respiratory distress either did not occur or did occur but was not known to the authors. Notably, all but one of the fourteen respiratory distress cases were associated with preterm birth (range: 34–36 weeks), indicating that respiratory distress occurred in at least 28% (13/46) of preterm births in MVID.

#### 3.3.3. Cholestasis

A life-threatening extra-intestinal symptom of MVID is intrahepatic cholestasis [21,22,23]. Intrahepatic cholestasis manifests in ~50% of all MVID patients [21]. The etiology of cholestasis in MVID is not clear, but may be caused by the *MYO5B* mutations carried by the patient and/or may be a complication of the TPN [21,22,23,24]. Although preterm birth in general is linked to a higher risk of neonatal hyperbillirubinia and jaundice due to liver immaturity, the percentage of MVID patients with reported liver disease or jaundice was similar in the preterm and term birth groups (liver disease reported in 55% (12/22) and 45% (10/22) of preterm and term births, respectively).

#### 3.3.4. Mortality

We next addressed mortality. Of the documented deaths in the MVID cohort, preterm birth (*t* (38) *=* −3.26, *p* = 0.0329), but not birth weight (*r* (27) = 0.1499, *p* = 0.46) or stool output (*r* (34) = −0.045, *p* = 0.79), was associated with a shorter average survival time (6.25 months for preterm births versus 16.31 months for at term births) (Figure 3B).

## 4. Discussion

This retrospective study revealed moderate to late preterm birth in more than half of all reported MVID cases. This is ~5-fold higher when compared to the world average (~10%) [25]. On a cautionary note, this number is based on cases for which gestational age was reported (50% (88/176) of all cases), and a reporting bias cannot be excluded. Nonetheless, if in the unlikely scenario that in all cases where gestational age was not reported patients would have been born at term, the percentage of preterm patients would be at least 26% (46/176), which is still significantly higher than the world average. MVID fetuses thus showed an increased odds of being born preterm.

Moderate to late preterm birth is generally associated with higher rates of infant morbidity and mortality when compared to full term birth [26]. A higher average stool volume in MVID was observed in the preterm birth group when compared to the at term birth group. Furthermore, a reduction in the average survival time was observed in the preterm birth group when compared to the at term birth group. Hence, moderate to late preterm birth appeared as a contributing factor with regard to MVID morbidity and mortality.

The gastrointestinal system is subject to rapid growth and maturation in the final weeks before birth. Consequently, moderate to late preterm birth is associated with immaturity of the gastrointestinal system. Preterm birth can be expected to contribute to, aggravate or complicate gastrointestinal symptoms in MVID neonates.

Neonatal respiratory distress was noted in at least one third of all reported moderate to late preterm MVID births. The risk of respiratory distress associated with preterm birth in general is well known, but it occurs predominantly in early preterm birth and much less in moderate to late preterm birth (declining from 98% to 5% in those born at 24 and 34 weeks of gestation, respectively [27]). The relatively high percentage of respiratory distress reported in moderate to late preterm MVID patients may therefore suggest that preterm MVID patients were at an increased risk of developing neonatal respiratory distress.

Why MVID patients were at risk of preterm birth is an outstanding question. There was no correlation between preterm birth in MVID and reported maternal risk factors or complications during pregnancy. Antenatal symptoms [28] occurred in almost one-quarter of MVID cases, but were not different between the preterm and at term birth groups. Increased fetal growth has been shown to increase the risk of spontaneous late preterm birth [29,30]. Higher birth weight percentiles were observed in MVID patients who were born preterm, when compared to those born at term. Whether this was specific for MVID or a more general phenomenon associated with late preterm birth [29] remains to be determined.

Several maternal factors associated with preterm birth risk, such as maternal age, size or smoking behavior were not typically reported in published case studies. However, there is no reason to believe that these factors were unevenly distributed between parents of MVID patients and non-MVID patients. It is also unlikely that the high percentage of preterm birth in MVID was related to the patients’ country of origin, as the range of average preterm birth percentages between high- and low-income countries is 9–12%, respectively, and none of the reported MVID patients originated from world regions with the highest preterm birth rate [25].

MVID belongs to a subgroup of congenital diarrheal disorders [13,31]. These include, in addition to MVID, tufting enteropathies and trichohepatoenteric syndrome (THES). Caralli and colleagues reported a high percentage (50%, 14/28) of preterm birth in THES [13]. In contrast to MVID, THES was independent of gestational age, associated with very to extremely low birth weight percentile, indicative of intrauterine growth restriction (IUGR) [13]. In addition to THES, congenital chloride losing diarrhea caused by *SLC26A3* mutations was associated with very (29%) and moderate to late (59%) preterm birth, polyhydramnios and IUGR [32]. While loss of the SLC26A3 protein has been reported in the MVID intestine [33], MVID was not associated with IUGR but rather with increased fetal growth. Other congenital diarrheal disorders and enteropathies associated with enterocyte defects were not associated with preterm birth [13], suggesting that prenatal intestinal dysfunction as such does not appear to predispose to preterm birth.

Possibly, some *MYO5B* mutations may be independent risk factors for preterm birth. Little is known about genes associated with preterm birth [34]. Preterm birth was restricted to MVID patients with *MYO5B* mutations. We found no link between *MYO5B* and other genes associated with diseases with preterm birth as a phenotype (HP:0001622; Human Phenotype Ontology (https://hpo.jax.org; accession date: 30 August 2021). Interestingly, *EBF1*, which encodes a transcription factor for—among many others—*MYO5B*, was identified in a genome-wide association study searching for loci associated with gestation age and preterm birth [35]. As timing of delivery involves concerted actions of the fetus, uterus, decidua and placenta, it would be of interest to investigate the effect of fetal and/or maternal mutations in the *MYO5B* gene on embryonal and fetal development, and on the function of the placenta in which *MYO5B* is highly expressed (Human Protein Atlas: https://www.proteinatlas.org/ENSG00000167306-MYO5B/tissue/placenta; accession date: 30 August 2021). Such studies may contribute to a better understanding of genetic and cell biological mechanisms that can lead to preterm birth.

In conclusion, we demonstrated a high percentage of idiopathic preterm births in *MYO5B*-associated MVID, and, moreover, put this finding in a clinical perspective with regard to MVID morbidity and mortality. Preterm birth and therewith associated adverse effects should thus be considered in the care of children diagnosed with MVID. Documentation of gestation age may contribute to a better prognostic risk assessment in MVID.

## Figures and Tables

**Figure 1 jcm-10-03935-f001:**
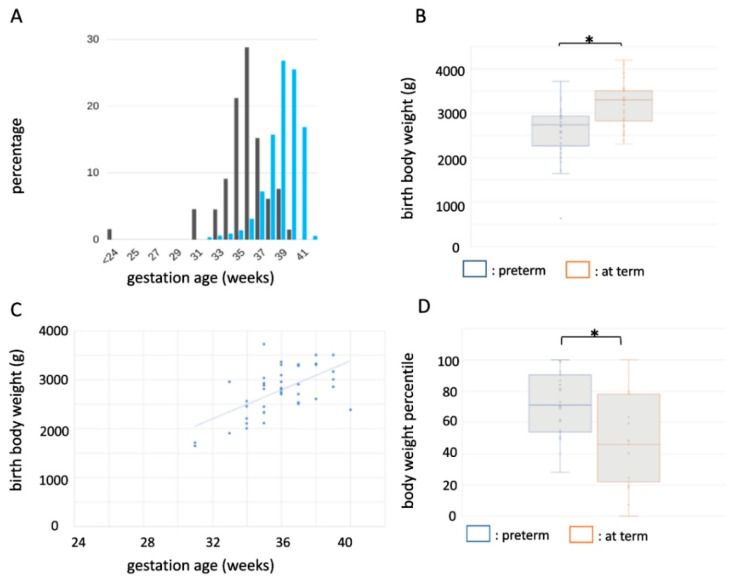
Preterm birth characteristics of MVID. (**A**) Distribution of gestational ages in MVID (black bars) versus population (blue bars). (**B**) Box-and-whisker plot showing average and mean body weights in MVID patients born preterm (blue) and at term (orange). (**C**) Correlation between gestational age and body weight in MVID. (**D**) Box-and-whisker plot showing average and mean body weight percentiles in MVID patients born preterm (blue) and at term (orange). *: *p* < 0.05. For exact *p* values see text.

**Figure 2 jcm-10-03935-f002:**
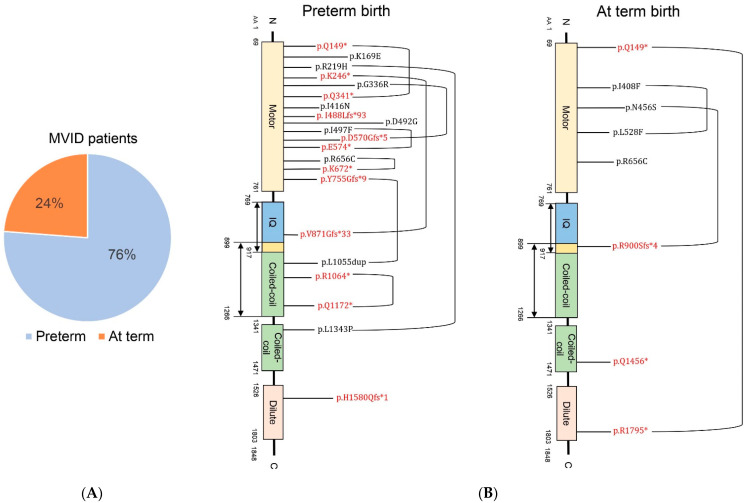
Relationship between *MYO5B* mutations and preterm birth in MVID. (**A**) Pie chart showing the percentage of preterm (blue) and at term (orange) births in cases of MVID associated with biallelic *MYO5B* mutations. (**B**) *MYO5B* mutations associated with preterm and at term births in MVID and the position of these mutation in the myosin Vb protein. Mutations that give rise to a premature termination codon are shown in red; other mutations are shown in black. Brackets connecting mutations indicate that these mutations were identified in the same patient. The absence of a bracket indicates a homozygous mutation. Asterisks indicate premature truncation of the protein.

**Figure 3 jcm-10-03935-f003:**
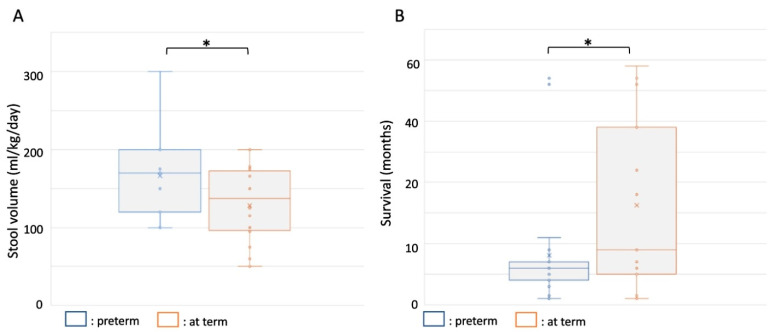
Stool volume and survival in MVID patients born preterm and at term. (**A**) Box-and-whisker plot showing average and mean stool volumes in MVID patients born preterm (blue) and at term (orange). (**B**) Box-and-whisker plot showing average and mean survival in MVID patients born preterm (blue) and at term (orange). *: *p* < 0.05 (for exact *p* values see text).

## Data Availability

Data supporting reported results are published with this article.

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
