# Peer review of "Risk and Clinical Significance of Idiopathic Preterm Birth in Microvillus Inclusion Disease"

_jcm, 2021, doi:10.3390/jcm10173935_

Round 1

Reviewer 1 Report

I believe that this manuscript should be strongly considered for publication by your journal. It is clearly written and the goals and objectives of the study are appropriately described as are their findings. This adds to the broader literature of this rare CODE disorder and its of important clinical significance. The most significant problem with the manuscript is table 1 which is largely unreadable and of minimal value. At a minimum, they should move it to the supplement but generate a meaningful diagram might be a better option. 

Author Response

We thank the reviewer for the positive response to our manuscript.

In the revised manuscript Table1 is now replaced with a more meaningful figure (figure 2 in the revised manuscript) that illustrates the MVID-linked MYO5B mutations associated with preterm and at term birth. 

Reviewer 2 Report

Microvillus inclusion disease (MVID) is an autosomal recessive disorder of intestinal epithelial cells and is characterized by intractable life-threatening watery diarrhea during infancy.

The authors conducted a comprehensive retrospective study to investigate factors influencing MVID morbidity and mortality. As a result, they demonstrated a high percentage of preterm births in cases of MYO5B-associated MVID.

This is an interesting report that represents a useful addition to the literature on this topic. Moreover, the paper is well-written and the statistical methods seem appropriate. However, I have the following concerns regarding the contents of the manuscript:

Minor comments:

  1. In the present study, 58% of preterm MVID cases showed a normal range (2500–4000g) of the average birth weight. In the discussion section, the authors stated that MVID was associated with increased fetal growth. Moreover, some previous studies demonstrated accelerated fetal growth patterns among pregnancies with preterm birth. What is the authors’ speculation of the mechanism of increased fetal growth in cases of MVID?

  1. In this study, preterm birth was associated with a shorter average survival time. The authors suggested that prematurity might be associated with the mortality. Indeed, previous studies have demonstrated that late preterm infants show higher rates of morbidity and mortality compared to full-term infants. However, it is possible that a fetus with serious MVID with a short predicted survival time after birth may be born prematurely. What is the authors’ opinion about this?

  1. In this article, the terminology “gestation age” is used in several sections, but this is commonly referred to as “gestational age.” This should be corrected.

Author Response

We thank the reviewer for the positive comments on our manuscript and valuable questions. 

  1. This is a very intriguing question. An earlier IVF study reported that high birth weight was associated with a higher number of inner mass cells (ICM; from which the fetus develops) versus throphoectoderm cells (TE; which forms the placenta) in the early embryo (PMID 25497449). The ICM/TE ratio is among others determined by the regulated orientation of cell divisions in the 8-cell stage embryo (PMID 20308546). As loss of MYO5B has been demonstrated to deregulate cell division orientation (PMID 31682603), it is tempting to speculate that the loss of MYO5B increases the chance of an embryo with a higher ICM/TE ratio and, consequently, increases fetal growth. We like to emphasize however that this is purely speculative and therefore should not be included in this manuscript. Instead we have argued that further investigation into the role of MYO5B in the developing embryo/fetus is warranted.
  2. We found no obvious differences between the seriousness (that is, time of onset of symptoms, TPN dependency) of MVID and whether patients were born  preterm or at term. Also, we found no correlation between the presence of antenatal symptoms and preterm birth. It appears that the classical  symptoms of MVID (secretory diarrhea, malabsorption, failure to thrive) start only after birth. Taking into account that other severe congenital diarrheal diseases such as tufting enteropathy are not typically associated with preterm birth, we consider it possible that loss of MYO5B affects processes during fetal development that affect the timing of birth and birth weight independently of the intestinal dysfunction (MVID). 
  3. We have corrected this throughout the manuscript.

Reviewer 3 Report

This study focused on the gestational age of cases diagnosed with MVID and explored the significance of this. The findings arising are helpful for individuals diagnosing and caring for individuals with MVID

SPECIFIC COMMENTS

  1. Please amend the term "MVID cases" or "MVID patients" to read "cases with MVID" or "patients with MVID"
  2. Some paragraphs of the MS include just one sentence. Please avoid single sentence paragraphs
  3. gestation age should be gestational age
  4. The comment about birth weight and birth weight centiles in the ABSTRACT is confusing. Clearly and intuitively if one is born prematurely one will be smaller. Please revise this to be more clear here and elsewhere
  5. There is repetition in the MS. Please revise to eliminate 
  6. The INTRO could be shortened and made more focused. Similarly (and even more so) for the DISCUSSION
  7. The METHODS contains the numbers of the cases. While the source of these are appropriate to be included in the METHODS, please consider how much should be in the RESULTS section instead
  8. The SUPPLEMENTARY TABLE referred to in the METHODS has no title or legend to guide the reader
  9. A definition of prematurity should be in the METHODS not the RESULTS. It would not be required to define expected terms (e.g. polyhydramnios) 
  10. Similarly, the description of the types of genetic findings (page 3/RESULTS) should be earlier in the MS (not in the RESULTS)
  11. The first section of section 3.3 is confusing: please revise. Further, there are a number of additional awkward sentences: please review grammar and use of language throughout
  12.  

Author Response

We thank the reviewer for the positive comments on our manuscript. 

  1. We have corrected this throughout the manuscript.
  2. We have avoided single sentence paragraphs.
  3. This has been corrected.
  4. We have revised this for clarity.
  5. We have eliminated repetition throughout the manuscript.
  6. We have shortened and focussed the introduction and discussion (please see changes with track changes). 
  7. We have changed this according to the reviewer's suggestion. The methods are in the Methods section and the results (numbers) in the Results section.
  8. A title and legend have been added to the supplementary Table. 
  9. We have changed this according to the suggestion.
  10. This has been corrected.
  11. We have revised section 3.3 and corrected grammar and use of language throughout the manuscript. Please see changs made with track changes tool. 

Round 2

Reviewer 3 Report

Thank for your revisions, that have resulted in an improved MS